Definition and review on a category of long non-coding RNA: Atherosclerosis-associated circulating lncRNA (ASCLncRNA)

Lu Shanshan 1
Liang Qin 1
Huang Yanqing 1
Meng Fanming 22018112@csu.edu.cn 2
Liu Junwen liujunwen@csu.edu.cn 1 3
1 Department of Histology and Embryology, School of Basic Medical Science, Central South University , Changsha , Hunan Province , China
2 Department of Parasitology, School of Basic Medical Science, Central South University , Changsha , Hunan Province , China
3 China-Africa Research Center of Infectious Diseases, School of Basic Medical Sciences, Central South University , Changsha , Hunan Province , China
Ladomery Michael
Electronic publication date: 2020 Nov 11
Publication date: 2020
Volume: 8
Electronic Location ID: e10001
Received 2020 Mar 10; Accepted 2020 Aug 29
Copyright: ©2020 Lu et al.
Copyright year: 2020
Copyright holder: Lu et al.
License: This is an open access article distributed under the terms of the Creative Commons Attribution License, which permits unrestricted use, distribution, reproduction and adaptation in any medium and for any purpose provided that it is properly attributed. For attribution, the original author(s), title, publication source (PeerJ) and either DOI or URL of the article must be cited.
License URL: https://creativecommons.org/licenses/by/4.0/

Keywords: Atherosclerosis, Biomarker, Circulating lncRNAs, Non-invasive diagnosis

Funding: National Natural Science Foundation of China 81770462 Fundamental Research Funds for the Central Universities of Central South University 2020zzts781 This work was supported by the National Natural Science Foundation of China [grant number 81770462] and the Fundamental Research Funds for the Central Universities of Central South University [2020zzts781]. The funders had no role in study design, data collection and analysis, decision to publish, or preparation of the manuscript.

==============================
Atherosclerosis (AS) is one of the most common cardiovascular system diseases which seriously affects public health in modern society. Finding potential biomarkers in the complicated pathological progression of AS is of great significance for the prevention and treatment of AS. Studies have shown that long noncoding RNAs (lncRNAs) can be widely involved in the regulation of many physiological processes, and have important roles in different stages of AS formation. LncRNAs can be secreted into the circulatory system through exosomes, microvesicles, and apoptotic bodies. Recently, increasing studies have been focused on the relationships between circulating lncRNAs and AS development. The lncRNAs in circulating blood are expected to be new non-invasive diagnostic markers for monitoring the progression of AS. We briefly reviewed the previously reported lncRNA transcripts which related to AS development and detectable in circulating blood, including ANRIL, SENCR, CoroMarker, LIPCAR, HIF1α-AS1, LncRNA H19, APPAT, KCNQ1OT1, LncPPARδ, LincRNA-p21, MALAT1, MIAT, and UCA1. Further researches and a definition of atherosclerosis-associated circulating lncRNA (ASCLncRNA) were also discussed.

Introduction

Cardiovascular diseases (CVDs) seriously endanger human health around the world, taking the lives of around 17.9 million each year (WHO, 2017), which place a heavy financial burden on society and families. Atherosclerosis (AS) is the most common cause of CVDs (Sanada et al., 2018). Although it is an asymptomatic condition, the accumulation and rupture of atheromatous plaques in arteries can lead to serious consequences, such as coronary artery diseases (CADs), acute myocardial infarction (AMI), and heart failure (HF), etc (Harada et al., 2014; Libby, Ridker & Hansson, 2011). Plaque rupture is responsible for 75% of AMI with highest incidence occurs in male beyond age 45 and female beyond age 50 (Pahwa & Jialal, 2019). The AS progression involves dynamic changes in the vessel wall, such as endothelial dysfunction, macrophage activation, and phenotypic changes of vascular smooth muscle cells (VSMCs) (Xi et al., 2013).

Compared with other morbid states, AS is characterized by small lesions and insidious onset. At present, the diagnosis of AS is based on a combination of multiple detection methods, such as echocardiography, electrocardiogram, computed tomography scan, blood test, and angiography (NHLBI, 2020). However, the confirmed diagnosis of AS depends on surgery and pathological examination (like carotid endarterectomy), which can be traumatic and risky for patients. Considering the difficulty of detection on such subtle changes in vivo, identifying potential biomarkers associated with the complex pathological progress of AS is essential for the prevention and treatment of AS.

Long noncoding RNAs (lncRNAs) are defined as transcripts > 200 bp without protein-coding potential. It was previously considered to be the “noise” or “junk” of the genome and have no substantive function (Palazzo & Lee, 2015). Recent researches, however, found that lncRNAs functions in various kinds of cellular activities, which are related to many serious human diseases (Wapinski & Chang, 2011). LncRNAs can target microRNAs (miRNAs) to form the competing endogenous RNA (ceRNA) axes and then function in different stages of AS development (Li, Zhu & Ge, 2016; Salmena et al., 2011). They not only exist in cells but can also be detected in plasma or serum samples, which are called circulating lncRNAs (Cao et al., 2019; Wang et al., 2017). Circulating lncRNAs show great resistance to endogenous RNase, which makes them more stable in blood samples (Tong et al., 2015). All of these provide the necessary basis for finding circulating lncRNA as potential biomarkers in AS prevention and treatment.

In this review, we briefly discuss current advances of circulating lncRNAs in AS, which aim to facilitate our understanding of the relationships between them and promote the development of circulating lncRNAs as predicting AS biomarkers for clinical applications.

Survey Methodology

Our team put the focus on advances in the relationship between epigenetics and CVDs. We performed the literature search mainly using the PubMed database (https://www.ncbi.nlm.nih.gov/pubmed/), Web Of Science (http://www.webofknowledge.com), and Google scholar (scholar.google.cn). Based on the main keywords-“atherosclerosis”, “coronary artery disease (or CAD)”, “heart failure” and “stroke” combined with “lncRNA”, “blood”, “serum”, or “plasma”, “exosome”, “microvesicle”, “apoptotic body”, relevant articles are extracted to classify and summarize the potential atherosclerosis-associated circulating lncRNAs. According to the search strategy, a total of 285 literatures were identified, including 55 reviews and 6 clinical trials. 282 of them have the full text. After reading the abstract and screening, 109 were then included which satisfied our goal. We did not refine factors such as journal, publishing date, or journal impact factors during our search. Ultimately, the time span of references in this review is from 1993 to 2020.

Circulating lncRNAs

Ideal biomarker refers to those molecules that can be used in non-invasive detection with relative stability, detection sensitivity, and specificity (Qi, Zhou & Du, 2016; Shi & Yang, 2016). But so far, lncRNAs used in scientific researches were usually derived from atherosclerotic plaques based on interventional methods (Roth & Diederichs, 2016). With the rapid improvements of detection methods, recent studies show that lncRNAs can be stably detected from body fluids (Panzitt et al., 2007; Reis & Verjovski-Almeida, 2012; Tinzl et al., 2004). LncRNAs can be secreted from tumor cells, then move into the circulatory system through exosomes, microvesicles, and apoptotic bodies, thus becoming a novel non-invasive diagnostic marker for monitoring the progression of cancers (Wang et al., 2019a). The variation of cellular components in blood-vessel cells during the development of AS make it possible for lncRNAs to migrate into the circulatory system (Monteiro et al., 2019).

Blood is the most widely distributed body fluid, carrying oxygen, nutrients, and signaling molecules to tissues and organs throughout the body. The circulating blood emphasizes a state of blood fluid distributing in the circulatory system of human body. Dynamic changes of particular substance in the blood may be closely related to specific disease states, such as cancers, CVDs, and nervous system diseases (Chen et al., 2017; Fridman et al., 2017; Wang et al., 2018). Such biomarkers can provide the following information: (1) identifying and classifying patient’s condition; (2) diagnosing and monitoring disease states; and (3) guiding doctors to make appropriate therapeutic schedules and prognosis observations (Heil & Tang, 2015). The properties of lncRNAs determine their potential as biomarkers, such as lncRNA PCA3 in prostate cancer, and lncRNAs UCA1 in bladder cancer. Both of them can be easily detected in urine samples, and even have special discriminations for cancer types (Hessels et al., 2003; Wang et al., 2006). Later, lncRNAs with biomarker function in circulating blood were found in gastric cancer, lung cancer and breast cancer (Dong et al., 2015; Liang et al., 2016; Zhang et al., 2016).

AS is one of the most common diseases in the world. Studies on the correlation between circulating lncRNAs and AS development have been reported in recent years (Chi et al., 2017; Pan et al., 2019; Wang et al., 2017). If pathological changes of vessel can be found in the early stage based on the detection of circulating lncRNAs, it will be shade light on the initial intervention and treatment to reducing the morbidity and mortality of patients.

ASCLncRNA: atherosclerosis-associated circulating lncRNA

ANRIL (CDKN2B antisense RNA 1)

ANRIL, also known as CDKN2B-AS1, is located at chromosome 9p21 (Pasmant et al., 2007). The ANRIL gene contains 19 exons and can be transcribed into many spliceosomes with tissue specificity (Folkersen et al., 2009). Genome-wide association analysis (GWAS) revealed many disease-associated single nucleotide polymorphisms (SNPs) at the 9p21 locus (Pasmant et al., 2011). These SNPs are closely associated with the severity of coronary AS, cervical AS, peripheral arterial disease (Congrains et al., 2012; Holdt et al., 2010; Holdt & Teupser, 2012).

ANRIL exists in many AS-associated cells and tissues, such as VSMCs, endothelial cells (ECs), monocytes, macrophages, and carotid tissues. It plays a trans-regulatory role through binding to Alu elements in the promoter region of target genes, which is involved in the regulation of fatty acids, glucose metabolism, and inflammatory responses (Bochenek et al., 2013; Holdt et al., 2013).

In a sequencing analysis of whole blood samples from patients with AMI, researchers detected ANRIL’s significant reduction and selected it as a self-labeling molecule (Vausort, Wagner & Devaux, 2014). This is the first report on the dynamic expression level of ANRIL in circulating blood. Another research exhibits that ANRIL was up-regulated in peripheral venous blood from patients of type 2 diabetes mellitus complicated with AMI. Recently, circulating ANRIL was found to increase in the plasma of CAD patients compared with the controls and showed promise as a good diagnostic value for CAD because it could be related to the severity of inflammation, stenosis degree, and prognosis (Hu & Hu, 2019). The differences in research objects, such as arterial blood vs. venous blood, and the existence of comorbidity may be the core reason for the opposite results. Since the mechanism of ANRIL’s function in the development of AS is not very clear, its role as a biomarker in circulating blood remains to be explored.

SENCR (smooth muscle and endothelial cell-enriched migration/ differentiation-associated long noncoding RNA)

SENCR is a specifically cytoplasmic lncRNA enriched in vascular cells. It was firstly discovered in the high-throughput sequencing of human coronary artery SMCs (Bell et al., 2014). Bell and colleagues found SENCR seems to play a role in inhibiting phenotypic transformation and cell proliferation of VSMCs through downregulating the cell contraction-related gene (Myocd) and upregulating the promigration-associated genes (Mdk/Ptn) (Bell et al., 2014). It can inhibit the pathological migration of VSMCs to neointima during AS formation. Besides, SENCR could also bind CKAP4, a cytosolic protein in ECs, through a noncanonical RNA-binding domain. This interplay helps maintain the adherent junctions, membrane integrity, and permeability of ECs, thus protecting against AS (Lyu et al., 2019). The expression level of SENCR, which was isolated from the vessel wall ECs of patients with CVDs, was much lower than the control. It suggests that SENCR may be decreased in patients with endothelial dysfunction or AS (Boulberdaa et al., 2016). The decline of this atheroprotective lncRNA may provide a warning for the onset of AS.

By using amplification refractory mutation system-polymerase chain reaction, SENCR can be detected stably and sensitively in blood samples of patients with CAD (Shahmoradi et al., 2017). Besides, a large-scale investigation focused on the effect of pioglitazone on type 2 diabetes found that the expression of SENCR in the blood is related to the evaluation of diastolic function after drug treatment, and that its indication in drug treatment effect is better than other existing indicators (De Gonzalo-Calvo et al., 2016b).

CoroMarker (Aldo-keto reductase family 1 member B1 pseudogene 3)

Yang et al. screened 174 differentially expressed lncRNA transcripts in blood samples from CAD patients and healthy people through microarray analysis, then they further chose five candidate lncRNAs by improving screening criteria and verified them by quantitative Polymerase Chain Reaction (qPCR) technology. Receiver operating characteristic (ROC) curve analysis indicated that lncRNA AC100865.1 could meet the requirements of biomarkers and was named CoroMarker. Better sensitivity and predictive effects can be obtained when combined with other risk factors of CVDs (Yang et al., 2015). CoroMarker has now been demonstrated to be distributed in vesicles of circulating blood and monocytes. Knockdown of CoroMarker in THP-1 cells caused significant down-regulation of IL-1β, IL-6, and TNF-α, suggesting that CoroMarker may play an important role in the inflammatory response of AS (Cai et al., 2016b).

LIPCAR (mitochondrially encoded long non-coding cardiac associated RNA)

LIPCAR showed opposite expression trend before and after the AMI event; it was down-regulated at the early stage of AMI, but gradually increased in the subsequent period (Kumarswamy et al., 2014). Surprisingly, it was released from the mitochondrion in blood cells rather than cardiac cells, which may be the reason for the changes in LIPCAR expression in different stages of AMI (Schulte et al., 2019). Meng and colleagues investigated diagnostic value of 9 circulating lncRNAs in ST-segment elevation myocardial infarction and found that LIPCAR had better diagnostic accuracy than others (Li et al., 2018a).

In plasma, the expression of LIPCAR was inversely proportional to high-density lipoprotein cholesterol, suggesting it may lead to dyslipidemia, which is a key factor in the progression of AS (Zhang et al., 2017). Besides, overexpression of LIPCAR induced by the treatment of ox-LDL or platelet-derived growth factor BB could promote the proliferation, migration, and phenotypic transition of VSMCs, which proved the function of LIPCAR in the progression of AS (Wang et al., 2019c). The possibility of LIPCAR as biomarkers for CVDs was also confirmed by other studies (De Gonzalo-Calvo et al., 2016a; Santer et al., 2019). These may lay the theoretical basis for the use of LIPCAR as a biomarker for AS.

HIF1A-AS1 (HIF1A antisense RNA 1)

Brahma-related gene 1 (BRG1) is a central catalytic subunit in many chromatin-modifying enzymatic complexes and is involved in the regulation of gene expression through chromatin remodeling (Zhou et al., 2009). HIF1A-AS1 was found to be correlated with BRG1 in VSMCs detected by microarray in BRG1 gain- and loss-of-function experiment. The interaction between them regulates the proliferation and apoptosis of VSMCs in vitro (Wang et al., 2015c). The function of HIF1A-AS1 in VSMCs was confirmed by Xu et al. that its overexpression could inhibit cell proliferation (Xu et al., 2019). Silencing of HIF1A-AS1 could promote the proliferation and reduce the hyperlipidemia-induced apoptosis of human umbilical vein ECs (Wang et al., 2015b). The potential role of HIF1A-AS1 as a biomarker in circulating blood has already been verified in several cancers such as colorectal carcinoma and non-small cell lung cancer (Gong et al., 2017; Tantai et al., 2015). Both Zhao and Xu testified that expression of HIF1A-AS1 was dramatically increased in the blood of patients with aneurysm (Xu et al., 2019; Zhao et al., 2014). HIF1A-AS1 was also up-regulated in the exosomes extracted from the plasma sample of patients with AS, which was thought to be the result of the activation of VSMCs and ECs (Wang et al., 2017). These studies laid a foundation for the clinical application of HIF1A-AS1 in the AS diagnosis, but also ruled out its usage as a biomarker alone.

LncRNA H19 (H19 imprinted maternally expressed transcript)

LncRNA H19 is located near the insulin-like growth factor 2 (IGF2) gene in an imprinted region of chromosome 11 (Giannoukakis et al., 1993). The common polymorphisms of lncRNA H19 were related to the risk and hazard level of CAD in a Chinese population (Gao et al., 2015). By using short hairpin RNA (shRNA) in ox- LDL-treated Raw264.7 cells, Han demonstrated that miR-130b is an important target gene of lncRNA H19. The axis could regulate inflammation response and lipid metabolism, which could be used as a targeting site of treating AS (Han et al., 2018a). Another experiment conducted by Zhang indicated that lncRNA H19 acts as a ceRNA of miR-148b and thus modulates the WNT/β-catenin signaling pathway to promote proliferation and suppress apoptosis of ox-LDL-stimulated VSMCs (Zhang et al., 2018). Up to now, the expression level of H19 in the serum of patients with AS was all found to be up-regulated (Cao et al., 2019; Han et al., 2018a; Pan, 2017; Zhang et al., 2018).

APPAT (atherosclerotic plaque pathogenesis associated transcript)

The newly reported lncRNA APPAT also shows potential predictive function for AS progression. APPAT is a 669 bp intergenic long non-coding RNA containing four exons and localized in human chromosome 2. APPAT was found to be mainly distributed in the middle layer of the coronary vessel wall and located on the cytoplasmic region of VSMCs via immunofluorescence (Meng et al., 2018).

APPAT was discovered in blood samples. It declined inconspicuously in patients with angina pectoris (AP) through the method of case-control matching, whereas significantly down-regulated in AMI patients compared with the normal group. Further examination revealed a decrease in APPAT expression in tissues of severely stenotic coronary arteries. This detectable trend in blood, independent of disease factors such as diabetes and hypertension, has potential value for predicting and monitoring the development of AS (Meng et al., 2018).

KCNQ1OT1 (KCNQ1 opposite strand/antisense transcript 1)

Studies have shown that lncRNAs can affect the transcriptional activities of multiple genes by interacting with chromatin (Saxena & Carninci, 2011). The Kcnq1 locus spans more than 1 Mb on the chromosome and is located on the short arm of human chromosome 11 (11p15.5), containing 8-10 protein-coding genes and lncRNA KCNQ1OT1 (Paulsen et al., 1998). Through recruiting chromatin and DNA-modifying proteins, KCNQ1OT1 interacts with chromatin to form a complex folding structure and then silences multiple target genes in this region (Kanduri, 2011).

The expression of KCNQ1OT1 in atheromatous plaques was negatively correlated with the age of patients, laying the foundation for further revealing the association between age and atheromatous plaques (Arslan et al., 2017). Lately, it is also verified to be elevated in the peripheral blood monocytes of patients with CAD and blood samples of MI patients, which suggests its role in the diagnosis of AS (Zhang et al., 2019).

LncPPARδ (long noncoding peroxisome proliferator-activated receptor delta)

PPARδ belongs to the nuclear receptor superfamily (Giordano Attianese & Desvergne, 2015). Activation of PPARδ increases the cholesterol efflux of macrophages, inhibits the transmembrane migration activity of leukocytes or monocytes to the inner wall of arteries, and reduces the size of atheromatous plaques (Ehrenborg & Skogsberg, 2013). Cai and colleagues isolated a lncRNA transcript, NONHSAT112178, from the plasma of CAD patients and named it as LncPPARδ, which is located near the PPARδ. They further reported that the expression of PPARδ decreased significantly in THP-1 cells after the knock down LncPPARδ, indicating that LncPPAR δ may be involved in PPARδ-mediated inflammatory signaling pathway and play a role in the progression of AS and CAD. In the plasma test results, LncPPARδ expression is up-regulated in CAD patients and is shown to exist stably in the blood. When combined with factors such as gender and age, LncPPAR δ showed better predictive function for CAD patients (Cai et al., 2016a).

LincRNA-p21 (tumor protein p53 pathway corepressor 1)

LincRNA-p21 was firstly discovered in mice (Huarte et al., 2010). It interacts with heterogeneous nuclear ribonucleoprotein K (hnRNP-K), the repressive complex of p53, and then affects the expression of p53’s downstream target genes (Barichievy et al., 2018). Wu et al. found that it inhibits cell proliferation and helps bring about neointimal formation in damaged coronary arteries, thus affecting the progression of AS in mice. They also discovered that down-regulation of LincRNA-p21 in human VSMCs can also promote cell proliferation and inhibit apoptosis (Wu et al., 2014). Hu et al. found that overexpression of LincRNA-p21 shows the opposite trends in VSMCs (Hu et al., 2019a). These two consistent results suggested that LincRNA-p21 may play a protective role against AS. Cekin et al. (2018) found that LincRNA-p21 decreased about 7 fold in atherosclerotic coronary artery tissues compared with the normal arterials in the same individuals. The application of circulating LincRNA-p21 exhibits a well predictive power on brain carotid tumors, chronic hepatitis (Fayda et al., 2016; Yu et al., 2017), and thoracic aortic aneurysms (Hu et al., 2019a). Although there is no direct evidence indicating the link between circulating LincRNA-p21 with the progression of AS, it is reasonable to infer that circulating LincRNA-p21 could also work as a marker for identifying AS given its effect on VSMCs when combined with other diagnostic indicators.

MALAT1 (metastasis associated lung adenocarcinoma transcript 1)

MALAT1, also known as Neat2, was firstly demonstrated to be related to the metastasis of non-small cell lung cancer (Ji et al., 2003). It binds with polycomb 2 to regulate the proliferation of cells by relocating growth-control gene loci (Yang et al., 2011). Michalik et al. (2014) verified the significant function of MALAT1 in the balancing of the phenotype of ECs, which affects vascular growth in vivo. It could act as ceRNAs with miRNAs (such as miR-216a-5p and miR-155) to inhibit inflammatory cytokine release and promote cell autophagy in ECs (Li et al., 2018b; Wang et al., 2019b).

MALAT1 was also found to be expressed highly in the macrophages of rats with diabetic AS (Han et al., 2018b), the plasma of patients with acute cerebral infarction (Teng & Meng, 2019), and serum of AS patients (Wang et al., 2019b). This can be partially explained by the increase of MALAT1 expression in exosomes secreted by ECs (Gao et al., 2019). Surprisingly, the expression is downregulated in both human and mouse atherosclerotic plaques (Arslan et al., 2017; Cremer et al., 2019). We hypothesize that the high expression of MALAT1 in serum and exosomes originates from the cells surrounding pathological tissues and thus cause a decline in plaques. However, Li and colleagues found lower expression of MALAT1 in exosomes of ox-LDL-treated ECs compared with the normal ECs (Li et al., 2019). This difference suggests that the role of MALAT1 in AS remains unclear and deserves a multi-center investigation.

MIAT (myocardial infarction associated transcript)

MIAT was found in an SNP-rich region related to AMI. A small change in a single SNP locus can cause an up-regulation of MIAT expression levels (Ishii et al., 2006). Knockdown of MIAT in ECs can inhibit cell proliferation and migration in vitro (Yan et al., 2015). The expression of MIAT increased in blood samples of AS patients, while its target miR-181b was down-regulated. Further studies found that a ceRNA axis–MIAT-miR-181b-STAT3 –in aortic SMCs may participate in cell proliferation and apoptosis and then affect the occurrence and development of AS (Zhong et al., 2018). The same expression trend was verified in the AS mice model, where MIAT inhibits efferocytosis through targeting the miR-149-5p/CD47 to regulate plaque vulnerability (Ye et al., 2019). The MIAT-based ceRNA pattern provides a potential target for therapy of AS.

UCA1 (urothelial carcinoma associated 1)

UCA1 was originally identified as a highly specific and sensitive biomarker of bladder transitional cell carcinoma, which can be detected from urine (Wang et al., 2006). It could promote glucose metabolism, lactic acid production, cell proliferation, and inhibit apoptosis of bladder cancer cells (Li et al., 2014; Wang et al., 2008). The functions of UCA1 was later explored in other cells. For example, it could protect cardiomyocytes from H2O2-induced apoptosis by targeting miR-1(Yan et al., 2016). In cardiovascular diseases, Yin et al. found that the silencing of UCA1 could induce apoptosis and repress the viability, migration, tube formation of human microvascular ECs (Yin, Fu & Sun, 2018). Hu et al. found that the knockdown of UCA1 in THP-1 cells could repress the formation of foam cells and restrain the total cholesterol and triglyceride levels via sponging miR-206 (Hu et al., 2019b). Moreover, a stable presence of UCA1 was also detected in circulating blood and serum exosomes (Barbagallo et al., 2018; Wang et al., 2015a; Wang et al., 2006). The expression of UCA1 in plasma decreased in the early stage of AMI but gradually increased in the subsequent process (Yan et al., 2016). This phenomenon is very similar to the LIPCAR aforementioned (Kumarswamy et al., 2014). Given its good indicator role in cancer, we could expect the possibility of its clinical application in AS diagnosis and prognosis.

Conclusion

As a hotspot of non-coding RNAs, lncRNAs have attracted the attention of scholars because of their unique structural characteristics and functions. LncRNAs widely distribute in various organs, tissues, and cells, and have important roles in different types of CVDs, such as CAD, AMI, and so on (Huang, 2018). At present, the researches on lncRNAs in the occurrence and development of AS mainly focus on their effects on the lipid metabolism, aberrant proteolysis and cell activities such as impaired function of ECs, modulation of VSMCs’ phenotype, recruitment of inflammatory cells, the polarization of macrophages and formation of foam cell (Fasolo et al., 2019; Li, Zhu & Ge, 2016; Zhou et al., 2016). In the last several years, the advance of high throughput sequencing technology brought about abundant novel transcripts with or without function annotation. Researchers have classified lncRNAs into different types according to their sequence characteristics, locations on chromosomes, or functions. These classifications could provide brief information on their physical and chemical property. Besides, classification based on its role in biological or pathological processes could facilitate their further researches, especially for those correlated with a specific disease (Jarroux, Morillon & Pinskaya, 2017). For example, SAL-RNAs was classified as Senescence-associated lncRNAs (Abdelmohsen et al., 2013), and PCA3/PCAT1 as Prostate cancer-associated transcripts (PCATs) (Mitobe et al., 2018). We suggest clustering the lncRNAs reviewed in the present paper into a category of Atherosclerosis-associated circulating lncRNA (ASCL ncRNA). Accordingly, the ASCL ncRNA should meet the following features: (1) human-sourced long non-coding RNA transcript, (2) detectable in circulating blood, (3) expression level changes with disease development. This definition was limited in human beings because most lncRNAs sequence evolved rapidly and can’t be detected as homologues in the different animal models (Necsulea et al., 2014). Further, lncRNA transcripts in this classification have a potentially predictive value for monitoring AS progression. Because of the high incidence and severity of AS in human beings, looking for potential non-invasive diagnostic methods and detectable markers for disease prevention, early diagnosis, and treatment, as well as providing a reliable reference for prognosis and follow-up observations has become an urgent task. Due to the chronic progression of AS, it is also necessary to screen out markers at different stages of disease progression. And the ASCLnRNA transcripts would be candidates. A summary on the characteristics of these ASCLnRNAs is collated in Table 1. Their function, potential mechanisms in human AS-related cells and dynamic changes in the circulating blood was provided here (Figs. 1 and 2).

Table 1 Summary of atherosclerosis-associated circulating lncRNA (ASClncRNA).

ASCLncRNA	Official Full Name	Gene ID (NCBI/Ensembl)	Category	Location	Atherogenic/ atheroprotective	Disease type	sensitivitya	specificitya	PMID	
ANRIL	CDmfKN2B antisense RNA 1	100048912/ ENSG00000240498	antisense	9p21.3	atherogenic	AMI/AS	81.6%-90.2%	59.7%-65.7%	30234067; 31411246; 23861667	
SENCR	Smooth muscle and endothelial cell enriched migration/differentiation-associated lncRNA	100507392/ ENSG00000254703	antisense	11q24.3	atheroprotective	AS	–	–	30584103	
CoroMarker	Aldo-keto reductase family 1 member B1 pseudogene 3	729347/ENSG00000213785	intergenic	11p15.2	atherogenic	CAD	76%	92.5%	26857419	
LIPCAR	Mitochondrially encoded long non-coding cardiac associated RNA	–	antisense	–	atherogenic	CAD/HF	72.2%	62.3%	31603865; 28790415	
HIF1A-AS1	HIF1A antisense RNA 1	–	antisense	14q23.2	unknown	AS	–		24875884	
LncRNA H19	H19 imprinted maternally expressed transcript	283120/ ENSG00000130600	intergenic	11p15.5	atherogenic	AS	53.6%	73%	30778327; 28165553; 28790415	
APPAT	–	ENST00000620272	intergenic		unknown	AMI	78.72%	93.02%	29372117	
KCNQ1OT1	KCNQ1 opposite strand/antisense transcript 1	10984/ ENSG00000269821	antisense	11p15.5	unknown	AMI/CAD/AS	100%	60%	30941792	
LncPPAR δ	–	–	intergenic	–	unknown	CAD	70%–82%	78%–94%	26871769	
LincRNA-p21	Tumor protein p53 pathway corepressor 1	102800311	intergenic	6p21.2	atheroprotective	AS	–	–	25156994	
MALAT1	Metastasis associated lung adenocarcinoma transcript 1	378938/ ENSG00000251562	intergenic	11q13.1	atheroprotective	AS	50%	63.6%	30586743; 31188931	
MIAT	Myocardial infarction associated transcript	440823/ ENSG00000225783	intergenic	22q12.1	atherogenic	AS	0.955	0.727	31237148; 31188931	
UCA1	Urothelial cancer associated 1	652995/ ENSG00000214049	intergenic	19p13.12	atherogenic	AMI	–	–	30633352	
Notes.

a The data of specificity/sensitivity percentages were derived directly from the ROC analysis of the original text.

AMI acute myocardial infarction

AS atherosclerosis

CAD coronary artery disease

HF heart failure

Figure 1 Functional mechanisms of Atherosclerosis-associated circulating lncRNA (ASClncRNA).

ECs, endothelial cells; VSMCs, vascular smooth muscle cells.

Figure 2 The dynamic changes of Atherosclerosis-associated circulating lncRNA (ASCLncRNA) in blood.

The researches on ASCLnRNAs in circulating blood are limited because previous studies just describe the occurrence and variation characteristics of ASCLnRNAs in AS (Chi et al., 2017; Zhang et al., 2019). Before further application of lncRNAs as biomarkers, some questions should be explored and answered. First, current testing methods are still limited, laying the challenges for exploring the origin of lncRNAs in circulating blood. Second, research methods are lacking standardization, and the sample size is often small (Moldovan et al., 2014). Third, some ASCLnRNAs do not only exists in cells related to CVDs but also in some cancer cells, which reduces their potential as independent biomarkers. Last but not least, the expression level of lncRNAs is generally lower than protein-encoding genes, which poses a challenge to the large-scale screening for biomarkers with repeatability and reliability. A key factor for future research work is the standardization, including standardized sample extracting and processing methods. Through this way, we could easily compare the different researchers’ data and draw reliable conclusions (Kumar et al., 2019). The advent of the latest third-generation full-length transcriptome sequencing technology makes it possible to screen a large number of lncRNAs in different diseases (Mercer et al., 2011). Fortunately, we have seen similar work in cancer-related clinical practice. For example, lncRNA-PCA3 has been widely recognized as a non-invasive diagnostic marker for prostate cancer (Sanda et al., 2017). Such work provides guidance and reference for finding AS-related biomarkers. The transcripts that meet the criterion of ASCLnRNA could form a candidate repository for further screening and validating researches on biomarkers for AS and could even become new therapeutic targets (Skuratovskaia et al., 2019).

Abbreviations

AMI acute myocardial infarction

ANRIL CDKN2B antisense RNA 1

APPAT atherosclerotic plaque pathogenesis associated transcript

AS atherosclerosis

ASCLncRNA atherosclerosis-associated circulating lncRNA

BRG1 Brahma-related gene 1

CAD coronary artery disease

ceRNA competing endogenous RNA

ECs endothelial cells

HF heart failure

HIF1A-AS1 LncRNA HIF 1 alpha-antisense RNA 1

LncPPARδ long noncoding Peroxisome proliferator-activated receptor delta

lncRNAs long noncoding RNAs

MALAT1 Metastasis-associated lung adenocarcinoma transcript 1

MIAT myocardial infarction associated transcript

miRNAs microRNAs

qPCR quantitative Polymerase Chain Reaction

SENCR smooth muscle and endothelial cell-enriched migration/ differentiation-associated long noncoding RNA

SMCs smooth muscle cells

SNPs single nucleotide polymorphisms

UCA1 urothelial carcinoma associated 1

VSMCs vascular smooth muscle cells

Additional Information and Declarations

Competing Interests

Author Contributions

Data Availability

The authors declare there are no competing interests.

Shanshan Lu performed the experiments, analyzed the data, prepared figures and/or tables, authored or reviewed drafts of the paper, and approved the final draft.

Qin Liang and Yanqing Huang performed the experiments, authored or reviewed drafts of the paper, and approved the final draft.

Fanming Meng and Junwen Liu conceived and designed the experiments, performed the experiments, analyzed the data, authored or reviewed drafts of the paper, and approved the final draft.

The following information was supplied regarding data availability:

There is no raw data in this review.

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
