# Peer review of "Definition and review on a category of long non-coding RNA: Atherosclerosis-associated circulating lncRNA (ASCLncRNA)"

_PeerJ, doi:10.7717/peerj.10001_

## Round 0.1 · original submission · Major Revisions

Dear Author,

As you will see, one of the reviewers raised a number of major points that I would like you to address. Furthermore, I would like to add that your review would benefit from some illustrative figures.

With best regards

Dr. Michael Ladomery - Editor

·

Basic reporting

1. Language is well.
2. Literature well referenced & relevant.
3. This field (Atherosclerosis-associated circulating lncRNA) was not been reviewed recently.

Experimental design

1. Article content is within the Aims and Scope of the journal.
2. This review organized logically into coherent subsections.

Validity of the findings

No comment.

Additional comments

Atherosclerosis (AS) is one of the most common reasons of cardio-cerebral-vascular system diseases, which threaten human health in modern society. Considering the difficulty of detection on such subtle changes in vivo, potential biomarkers during the complex pathological progress of AS is of great significance for the prevention and treatment of AS. Long noncoding RNAs (lncRNAs) are defined as transcripts >200 bp which commonly have no protein coding potential. LncRNAs can be secreted into the circulatory system through exosomes, microvesicles and apoptotic bodies. The lncRNAs in blood are expected to be new non-invasive diagnostic markers for monitoring the progression of AS. In this manuscript, Dr. Shanshan Lu and colleagues briefly reviewed the previously reported lncRNA transcripts which related to AS development and were detectable in circulating blood. Although the current article is interesting, there are some minor concerns that need to be addressed before consideration of publication.

Minor comments:
1. To avoid confusion caused by the name. All LncRNAs should have uniform code from NCBI (https://www.ncbi.nlm.nih.gov) or Ensembl (http://asia.ensembl.org/index.html).
2. Whether the mentioned LncRNAs interact with each other or have synergistic effects on some signaling pathways. It is suggested that the molecular mechanism of lncRNA should be included in the schematic diagram.
3. Can the diagnostic efficacy of the mentioned LncRNAs be compared in a table?

·

Basic reporting

The review's introduction has not been adequately introduced. Additional information on the criticality of AS, it's incidence & mortality rates, etc. may be elaborated upon. The authors have also introduced the term "circulating lncRNAs" (line 87) but have not clarified on its origin at the juncture. Additional review of the available literature is recommended to draft a more explanative introduction section.

Citation not provided in many parts of the manuscript. The authors must revise their article with proper references added wherever necessary. Some examples of missing citations include Lines 106-109, 110, 128, 135.

What do the authors mean by "Circulating blood"?

Experimental design

In the search terms used, why was CAD included when it is only a further complication of AS? If so, why were other terms like stroke and HF not included?

What was the time-span of the search? Papers published from which year until which year were included in this review?

Validity of the findings

The authors have collated information on "circulatory lncRNAs" involved in AS through literature mining on PubMed. The findings have not been summarized in a simple, cohesive manner.

The role of ncRNAs in AS has been reviewed and published quite recently in an elaborate manner (Refer: https://doi.org/10.1093/cvr/cvz203). However, this paper hasn't even been cited in this review. The only novelty in this manuscript is the focus on "circulating" lncRNAs - however, the literature survey done by the authors seems inadequate and the manuscript is also very poorly written. Other recent literature on AS & lncRNAs which are notable in their exclusion include PMIDs 30384894, 30300747 and 31212708 to name a few.

The authors also suggest clustering the lncRNAs mentioned in this paper as ASCLncRNAs - however, there does not seem to be a specific need for this classification. In addition, some of the possible members like ANRIL and LIPCAR are also associated with other CVDs and cancers. Therefore, the classification seems redundant. The authors may perhaps choose to create an online repository (or) portal for the collection of such AS related circulating lncRNAs and this would serve a better purpose for the researchers in this domain.

Additional comments

This manuscript by Lu et al. presents a distinct set of lncRNAs related to atherosclerosis (ASCLncRNA). The work has been presented well. However, there are a few parameters which the authors may further fine-tune as suggested here.

The manuscript will greatly benefit from being reviewed by a native English speaker or a language editor. While small sections of the article have been written well, most parts seem repetitive/redundant and need to be further worked upon.

The authors may kindly provide the statistics about how many articles they retrieved from PubMed using the search terms - and info regarding any further filtration & inclusion/exclusion criteria. A simple graph depicting the growth of publications in this field may also be included.

Figure 1 may be amended to use a variation of colours - some of the arrows inside the cells and the text (like MALAT1) are not clearly visible. It would be better if the labelling was done externally.

---

## Round 0.2 · Minor Revisions

Dear authors, as you will see, whereas one of the reviewers was happy with the revision, the other reviewer still requests several changes, but these are minor in nature. I would ask you to address each of the points, and also please could you ensure that the English grammar is also perfected, thank you for understanding- the Editor.

·

Basic reporting

No comment.

Experimental design

No comment.

Validity of the findings

No comment.

Additional comments

Atherosclerosis (AS) is one of the most common reasons of cardio-cerebral-vascular system diseases, which threaten human health in modern society. Considering the difficulty of detection on such subtle changes in vivo, potential biomarkers during the complex pathological progress of AS is of great significance for the prevention and treatment of AS. Long noncoding RNAs (lncRNAs) are defined as transcripts >200 bp which commonly have no protein coding potential. LncRNAs can be secreted into the circulatory system through exosomes, microvesicles and apoptotic bodies. The lncRNAs in blood are expected to be new non-invasive diagnostic markers for monitoring the progression of AS. In this manuscript, Dr. Shanshan Lu and colleagues briefly reviewed the previously reported lncRNA transcripts which related to AS development and were detectable in circulating blood. The paper is improved and most concerned raised by the reviewer have been addressed. I think it is might suitable for publication at this version of revised manuscript.

·

Basic reporting

Line 82: AS is being compared to tumors, but the context here is not explained clearly.

Line 94: Kindly add a relevant reference for this statement.

Line 95-96: The authors mention the extreme temp. resistance qualities of lncRNA. However, they do not explain it's relevance in this context.

Line 119-122: The intended meaning is lost within the long sentence. The authors may kindly look into it and perhaps break it down into shorter, comprehensible sentences.

Line 144: "firstly discovered" might indicate it being found there first and later at a different location. Kindly make sure to fix this issue.

Lines 156-158, 190-191, 248-250, 377-378: Citation not provided.

Line 208: Plasma?

Experimental design

Line 111: The authors can add the points mentioned in the rebuttal letter about no. of manuscripts considered (285) and how full-texts were taken, in this main article as well. It would be a valuable addition to information.

The authors are also requested to kindly go through their paper in-depth and add citations wherever applicable - it is missing in many places!

The efforts of the authors to create the unified table is laudable. It would be more beneficial if they also add a note on how specificity/sensitivity percentages were calculated.

Validity of the findings

Line 165-166: Since the role of ANRIL in AS is not clear to date, does it ratify inclusion in this list?

In the case of some lncRNAs mentioned here, the authors have tried to combine findings from multiple studies, to propose a possible role for the lncRNAs - i.e., a lncRNA with a presence in circulation and having a separate role in disease progression. While the approach is commendable, the hypothesis will bear more credence when associated with some experimental proof. Therefore, these results seem hypothetical at best. And, it is for this reason, that it seems premature to me to classify these lncRNAs as ASCLncRNAs as suggested when sufficient experimental proof has not been put forth.

Additional comments

This manuscript by Lu and colleagues is a much-improved version of the previous draft. However, as mentioned above, there still remain some concerns to be addressed, as mentioned above. The authors and the manuscript would also benefit greatly from being reviewed once by a copy-editor or a native speaker after they complete making changes to it, to remove some grammatical errors.

The authors can add the note on circulating blood, as specified in the rebuttal, into the main manuscript to enable new readers to easily understand the paper better.

---

## Round 0.3 · accepted · Accept

Thank you for further improving the manuscript in response to reviewer 2.